# Diagnostic and Prognostic Potential of *SH3YL1* and *NOX4* in Muscle-Invasive Bladder Cancer

**DOI:** 10.3390/ijms26093959

**Published:** 2025-04-22

**Authors:** Mingyu Kim, Euihyun Jung, Geehyun Song, Jaeyoung Joung, Jinsoo Chung, Hokyung Seo, Hyungho Lee

**Affiliations:** Center for Urologic Cancer, National Cancer Center, 323, Ilsan-Ro, Ilsandong-Gu, Goyang-Si 10408, Gyeonggi-Do, Republic of Korea; kmk63819@ncc.re.kr (M.K.); jeh0315@ncc.re.kr (E.J.); ghsong@ncc.re.kr (G.S.); urojy@ncc.re.kr (J.J.); cjs5225@ncc.re.kr (J.C.)

**Keywords:** AKI, bladder cancer, cisplatin, *SH3YL1*, *NOX4*

## Abstract

Bladder cancer, especially muscle-invasive bladder cancer (MIBC), poses significant treatment challenges due to its aggressive nature and poor prognosis, often necessitating cisplatin-based chemotherapy. While cisplatin effectively reduces tumor burden, its nephrotoxic effects, specifically cisplatin-induced acute kidney injury (AKI), limit its clinical use. This study investigates *SH3YL1* as a potential biomarker for bladder cancer progression and AKI. Plasma and urine *SH3YL1* levels were measured in bladder cancer patients undergoing cisplatin treatment, showing elevated baseline levels compared to controls, suggesting a link with bladder cancer pathology rather than cisplatin-induced AKI. Functional network and Gene Ontology (GO) enrichment analyses identified *SH3YL1*’s interactions with NADPH oxidase pathways, particularly *NOX* family genes, and highlighted its roles in cell adhesion, migration, and cytoskeletal organization—processes critical for tumor invasiveness. Notably, *SH3YL1* and *NOX4* expression were significantly higher in MIBC than in non-muscle-invasive bladder cancer (NMIBC), with a strong correlation between *SH3YL1* and *NOX4* (r = 0.62) in MIBC, suggesting a subtype-specific interaction. Kaplan–Meier survival analysis using The Cancer Genome Atlas bladder cancer (TCGA-BLCA) data further demonstrated that low *SH3YL1* expression is significantly associated with poor overall and disease-specific survival in MIBC patients, reinforcing its role as a prognostic biomarker. In conclusion, *SH3YL1* is a promising biomarker for identifying the invasive characteristics of MIBC and predicting patient outcomes. These findings underscore the importance of *SH3YL1–NOX4* pathways in MIBC and suggest the need for further research into targeted biomarkers for bladder cancer progression and cisplatin-induced AKI to improve patient outcomes in high-risk cases.

## 1. Introduction

Bladder cancer is a prevalent malignancy primarily originating in the bladder, which is part of the urinary tract and is pathologically classified into two major subtypes: NMIBC and MIBC [1,2,3,4,5,6,7,8]. NMIBC is generally confined to the mucosa and submucosa layers of the bladder wall, generally carries a favorable prognosis, and is managed with local treatments such as transurethral resection and intravesical therapy aimed at reducing recurrence [9,10,11]. In contrast, MIBC is a more aggressive form, characterized by its propensity to invade the muscle layer of the bladder, and is associated with a high metastatic potential and poor prognosis, often necessitating intensive systemic treatment [12,13,14]. For advanced or metastatic bladder cancer, cisplatin-based chemotherapy remains a mainstay, valued for its potent cytotoxic effects and efficacy in reducing tumor burden [15,16,17,18]. However, the clinical utility of cisplatin is significantly limited by its nephrotoxicity, primarily manifesting as cisplatin-induced AKI, a serious adverse effect that can hinder therapeutic outcomes and restrict its use in patients with pre-existing renal dysfunction [19,20,21,22]. The risk of AKI not only poses a substantial challenge in managing bladder cancer but also underscores an urgent need for reliable biomarkers to predict or monitor renal injury in patients receiving cisplatin. Also, various studies and guidelines on biomarkers to predict AKI are continually being reported [23,24,25,26,27]. Such biomarkers could enable more personalized treatment regimens by facilitating early intervention and optimizing the risk–benefit balance in cisplatin therapy. In recent years, *SH3YL1* has gained attention as a potential biomarker for AKI, with evidence suggesting its upregulation in various kidney injury models, positioning it as a promising candidate for early detection and monitoring of renal damage. Yoo et al. [28] demonstrated that SH3YL1 is significantly involved in kidney injury pathways, particularly through its interaction with NADPH oxidases such as *NOX4*, which is known to mediate reactive oxygen species (ROS) production in oxidative stress conditions. Similarly, Li et al. [29] reported that *NOX4* plays a critical role in AKI by exacerbating inflammation and cellular damage via ROS-driven mechanisms. These findings highlight the overlapping roles of *SH3YL1* and *NOX4* in kidney injury, further supporting their potential as biomarkers for cisplatin-induced nephrotoxicity *SH3YL1* [28,29,30,31,32]. While many studies have actively explored the link between *SH3YL1* and kidney injury, its expression and functional role in bladder cancer—particularly under cisplatin treatment—remain largely unexplored. Given cisplatin’s dual impact as both an antitumor agent and a nephrotoxin, understanding *SH3YL1*’s behavior in bladder cancer patients undergoing cisplatin therapy could reveal its potential as a biomarker for both tumor progression and renal injury. However, initial observations suggest that *SH3YL1* levels do not change significantly in blood and urine following cisplatin administration in bladder cancer patients, raising questions about its applicability as a universal AKI marker in this setting. This study aims to explore the expression patterns of *SH3YL1* in bladder cancer, with a particular focus on its correlation with *NOX4*, a gene associated with oxidative stress and implicated in cancer progression. Additionally, we examine how the invasive characteristics of MIBC and the non-muscle-invasive nature of NMIBC may differentially influence *SH3YL1* expression. By investigating these patterns, this study seeks to clarify whether *SH3YL1* can serve as a reliable biomarker for both kidney injury and bladder cancer progression, specifically in the context of cisplatin therapy.

## 2. Results

### 2.1. ELISA Analysis of SH3YL1 in MIBC Patients

To investigate *SH3YL1* as a potential biomarker for kidney injury, particularly in the context of cisplatin-induced nephrotoxicity, quantitative ELISA was conducted on plasma and urine samples from bladder cancer patients. The results revealed that, while *SH3YL1* levels in plasma did not significantly rise following cisplatin treatment, both plasma and urine *SH3YL1* levels were consistently higher in MIBC patients compared to healthy controls (Figure 1A,B). This elevation in baseline *SH3YL1* levels in patients, irrespective of acute treatment response, suggests that *SH3YL1* may reflect the underlying chronic pathology associated with MIBC rather than serve as an immediate marker for cisplatin-induced acute kidney injury. These findings indicate that while *SH3YL1* shows elevated levels in MIBC compared to healthy controls, its role as an acute injury marker, particularly for cisplatin-induced nephrotoxicity, remains uncertain. To further delineate *SH3YL1*’s involvement in MIBC pathology, additional analyses were performed to investigate its association with specific molecular pathways and cellular functions relevant to tumor progression and kidney injury.

### 2.2. Functional Network Analysis of SH3YL1-Related Genes Highlights NADPH Pathways

To elucidate the molecular network associated with *SH3YL1*, we utilized the STRING database, identifying 36 genes closely interacting with *SH3YL1*. These genes were grouped into three distinct clusters using k-means clustering based on their interaction patterns and functional similarities (Figure 2A). Pathway and process enrichment analysis using Metascape revealed significant associations with NADPH oxidase activity pathways, particularly involving *NOX* family members, as well as actin cytoskeleton dynamics, cell migration, and oxidative stress responses (Figure 2B). Details of the pathways and contributing genes associated with each node in the network are provided (Appendix A). GO enrichment analysis further highlighted the biological and functional relevance of *SH3YL1*-associated genes. The analysis revealed significant enrichment across three GO categories—biological processes (BP), cellular components (CC), and molecular functions (MF) (Figure 2C). The top enriched terms in the BP category included oxidative stress responses, cell adhesion, and cytoskeletal organization, while the CC analysis showed localization to the plasma membrane, actin cytoskeleton, and focal adhesion sites. The MF analysis identified enrichment in protein binding, kinase binding, and actin filament binding, underscoring the functional importance of *SH3YL1*-associated genes in regulating cellular motility and signal transduction. However, it is important to acknowledge that these findings are based solely on in silico analyses. While the results provide valuable insights into potential molecular mechanisms, further validation through experimental approaches, such as in vitro or in vivo studies, would be necessary to confirm these observations and establish causal relationships. These additional studies could better elucidate the specific roles of *SH3YL1* in cancer progression and its interactions with *NOX* family members.

### 2.3. SH3YL1 and NOX Family Gene Correlation Analysis in NMIBC and MIBC

To investigate the potential role of *SH3YL1* in oxidative stress regulation during bladder cancer progression, we focused on NADPH-related genes from the *SH3YL1* interaction network identified in STRING. These genes, including *NOX1*, *NOX4*, *NOX5*, *CYBA*, *NCF1*, and *NCF4*, were selected due to their established involvement in ROS generation and redox balance. An initial analysis of their relative expression levels in NMIBC and MIBC revealed distinct patterns, with *SH3YL1* and *NOX4* showing significantly higher expression in MIBC compared to NMIBC (Appendix A). This finding informed the subsequent correlation analysis, which aimed to uncover *SH3YL1*’s relationship with *NOX* family genes across these bladder cancer subtypes. Spearman’s correlation analysis revealed distinct patterns of association between the *SH3YL1* and *NOX* family genes in NMIBC and MIBC, suggesting potential shifts in oxidative stress regulatory mechanisms during cancer progression. In NMIBC, *SH3YL1* demonstrated weak correlations with most *NOX* family genes, including a slight positive association to *NOX5* (r = 0.11) and *NCF1* (r = 0.12), and a weak negative correlation with *NCF4* (r = −0.17). These findings indicate that *SH3YL1* does not have a strong coordinated expression with *NOX* family members in NMIBC, which might reflect its limited involvement in oxidative stress pathways at the non-invasive stage (Figure 3A). In contrast, MIBC displayed a marked shift in SH3YL1’s correlation profile, with a strong positive correlation observed between *SH3YL1* and *NOX4* (r = 0.62, *p* < 0.01). This suggests that *SH3YL1* and *NOX4* may act cooperatively in MIBC, potentially contributing to the invasiveness of cancer cells through enhanced oxidative stress modulation. Importantly, no other *NOX* family gene showed a statistically significant correlation with *SH3YL1* in MIBC. This highlights the unique role of the *SH3YL1–NOX4* axis in invasive bladder cancer.

### 2.4. ROC Curve Analysis for Evaluating SH3YL1 and NOX4 as Biomarkers in NMIBC and MIBC

To evaluate the potential of *SH3YL1* and *NOX4* as diagnostic biomarkers for bladder cancer subtypes, we conducted ROC (Receiver Operating Characteristic) curve analysis on *SH3YL1*, *NOX4*, and the combined *SH3YL1* + *NOX4* expression levels in NMIBC and MIBC groups. The area under the curve (AUC) values were used to quantify the predictive power of each marker and their combination, providing insight into their applicability in clinical settings. In the NMIBC group, *SH3YL1* showed an AUC of 0.605, while *NOX4* had an AUC of 0.604. The combined *SH3YL1* + *NOX4* expression resulted in an AUC of 0.496, indicating limited diagnostic utility within this subgroup (Figure 4A). These findings suggest that the biological roles of *SH3YL1* and *NOX4* may be more closely associated with the aggressive characteristics of MIBC rather than the less invasive NMIBC phenotype. The lower predictive values in NMIBC also highlight the specificity of *SH3YL1* and *NOX4* as potential markers for more invasive forms of bladder cancer. In the MIBC group, *SH3YL1* achieved an AUC of 0.730, while *NOX4* had an AUC of 0.750; both values demonstrated significant predictive power for differentiating MIBC cases (Figure 4B). The combined *SH3YL1* + *NOX4* expression yielded an AUC of 0.703, which, while slightly lower than the individual markers, still supports their complementary diagnostic value. The consistent predictive accuracy of *SH3YL1* and *NOX4*, individually and in combination, underscores their relevance as molecular indicators of the invasive phenotype associated with MIBC. Although the AUCs did not exceed 0.8, values above 0.7 indicate meaningful predictive power in identifying MIBC cases, even within the limitations of a small cohort.

### 2.5. Survival Analysis of SH3YL1 and NOX4 Expression in MIBC and NMIBC

To investigate the prognostic relevance of *SH3YL1* and *NOX4*, Kaplan–Meier survival analyses were performed using bladder cancer patients from the TCGA-BLCA dataset, which had been stratified into NMIBC and MIBC subgroups. Patients were dichotomized into high and low expression groups based on the median expression levels of each gene. The results of this analysis indicated that high *SH3YL1* expression was significantly associated with poor patient outcomes, particularly in the context of MIBC. In this subgroup, patients with low *SH3YL1* expression exhibited markedly reduced overall survival (OS) and disease-specific survival (DSS), with no individuals surviving beyond 20 months (OS: *p* = 9.665 × 10^−3^; DSS: *p* = 0.0224) (Figure 5). These findings underscore the significant prognostic value of *SH3YL1* in advanced-stage bladder cancer. Conversely, the prognostic significance of *SH3YL1* in NMIBC was less pronounced. Although low *SH3YL1* expression was associated with inferior OS (*p* = 0.0192), its association with DSS did not reach statistical significance (*p* = 0.0689), suggesting only a marginal role in early-stage disease (Appendix A). In contrast, the expression levels of *NOX4* exhibited a more variable pattern. In NMIBC, high *NOX4* expression exhibited a tendency towards improved survival, though this did not reach statistical significance (OS: *p* = 0.1734; DSS: *p* = 0.1876) (Appendix A). Conversely, in MIBC, there was no discernible correlation between *NOX4* expression and OS or DSS (OS: *p* = 0.6716; DSS: *p* = 0.6862) (Appendix A), suggesting its limited capacity to predict prognosis in isolation. However, when comparing patients across tumor subtypes, notable differences emerged. Specifically, among patients with high *NOX4* expression, those with NMIBC exhibited significantly better OS compared to MIBC (*p* = 0.0443) (Appendix A). Conversely, among patients with low NOX4 expression, those with MIBC had significantly worse DSS compared to NMIBC (*p* = 0.0002) (Appendix A). These observations suggest that the clinical impact of *NOX4* expression is context dependent, modulated by tumor stage and invasiveness. Collectively, these findings emphasize the significant prognostic value of *SH3YL1* in MIBC, and, to a lesser extent, in NMIBC. In contrast, *NOX4* appears to exert a more nuanced role, exhibiting subtype-specific influences that may reflect its involvement in the molecular heterogeneity of bladder cancer.

## 3. Discussion

Cisplatin remains a cornerstone chemotherapeutic agent in the management of advanced bladder cancer, particularly for MIBC and metastatic cases, due to its potent cytotoxic effects on rapidly proliferating tumor cells [12,33,34]. Despite its efficacy, cisplatin’s dose-limiting nephrotoxicity, specifically its propensity to induce AKI, poses significant challenges in clinical settings, often constraining its use and impacting patient outcomes due to renal impairment [35,36,37]. Recently, *SH3YL1* has emerged as a potential biomarker for AKI, particularly in contexts associated with increased oxidative stress, sparking interest in its expression dynamics in bladder cancer patients receiving cisplatin therapy [38]. Our analysis of *SH3YL1* expression in blood and urine samples of bladder cancer patients receiving cisplatin therapy revealed no significant changes post-treatment, suggesting that *SH3YL1* may not serve as a universal marker for cisplatin-induced AKI in this cohort. This lack of variation led us to examine *SH3YL1*’s role in bladder cancer progression, focusing on its associations with key oxidative stress-related genes such as *NOX4* across bladder cancer subtypes. This observation is aligned with previous reports showing that *NOX4* promotes epithelial–mesenchymal transition and metastatic progression via ROS-mediated mechanisms in various cancers [39]. This significant association reflects a potential adaptive mechanism linked to the aggressive and invasive characteristics of MIBC, highlighting the divergence in molecular pathways underlying NMIBC and MIBC pathophysiology. The pronounced *SH3YL1–NOX4* correlation in MIBC may reflect an upregulated oxidative stress response, potentially priming the invasive phenotype for enhanced survival, metastatic capability, or resistance to oxidative damage. Conversely, the absence of *SH3YL1–NOX4* correlation in NMIBC underscores the subtype-specific nature of this pathway, which may be less critical in non-invasive stages of bladder cancer. Interestingly, the absence of cisplatin-induced *SH3YL1* upregulation in MIBC patients could stem from the baseline activation of the *SH3YL1–NOX4* pathway, suggesting that further upregulation induced by cisplatin might be redundant. This contrasts with typical cisplatin-induced AKI mechanisms, which often involve acute oxidative stress activation. These findings imply that cisplatin-induced nephrotoxicity in MIBC may involve alternative molecular pathways beyond *SH3YL1* and *NOX4* activation, necessitating further studies to elucidate these mechanisms. Kaplan–Meier survival analyses highlighted *SH3YL1* as a strong prognostic biomarker, particularly in MIBC, where low expression was significantly associated with poor OS and DSS. In NMIBC, *SH3YL1* also showed a trend toward prognostic relevance, although the association with DSS did not reach statistical significance. This context dependency was further supported by inter-subtype comparisons, where MIBC patients with low *NOX4* expression exhibited significantly worse DSS compared to NMIBC counterparts. These findings suggest that *NOX4*’s prognostic role may be influenced by the tumor microenvironment and disease stage. Furthermore, ROC curve analysis highlighted the diagnostic potential of *SH3YL1* and *NOX4* for MIBC, with AUCs of 0.730 and 0.750, respectively. These values indicate moderate predictive accuracy, supporting their clinical utility for stratifying high-risk patients. However, the lower AUC values in NMIBC (0.605 for *SH3YL1* and 0.604 for *NOX4*) suggest limited diagnostic power within this less aggressive subgroup. This specificity in MIBC underscores the potential of *SH3YL1* and *NOX4* to identify patients who may benefit from intensified therapeutic interventions, paving the way for more personalized bladder cancer management strategies. Despite these promising findings, this study has limitations. The relatively small sample size necessitates further validation through larger cohorts, animal models, or in vitro studies. Additionally, pathway analysis exploring alternative mechanisms of cisplatin-induced AKI is warranted to comprehensively understand its nephrotoxic effects. Nonetheless, this study represents a significant step forward, being the first to examine the *SH3YL1–NOX4* relationship in MIBC patients undergoing cisplatin therapy. By bridging molecular insights with clinical outcomes, these findings lay the groundwork for targeted biomarkers and therapeutic strategies, addressing the critical need for precision in managing high-risk bladder cancer patients.

## 4. Material and Methods

### 4.1. Medical Samples

This study involved 80 participants, including 20 healthy individuals (control group) and 60 patients diagnosed with urothelial carcinoma (patient group). Among the patients diagnosed with bladder cancer, 16 patients were classified as MIBC patients, and 44 patients were classified as NMIBC patients based on pathologic classification (Appendix A). Written informed consent was obtained from all participants before sample collection, and the study protocol was approved by the National Cancer Center Institutional Review Board (NCC2022-0343).

### 4.2. Sample Collection and Processing

Blood samples were collected in EDTA tubes and plasma was separated by centrifugation at 1500× *g* for 10 min at 4 °C. Urine samples were collected in sterile containers and centrifuged at 1000× *g* for 10 min to remove debris. Both plasma and urine samples were aliquoted and stored at −80 °C until analysis. *SH3YL1* expression levels in plasma and urine were measured using enzyme-linked immunosorbent assay (ELISA) to assess the changes before and after cisplatin treatment.

### 4.3. Enzyme-Linked Immunosorbent Assay (ELISA)

Plasma and urine *SH3YL1* levels were quantified using a commercially available ELISA kit (MyBiosource, San Diego, CA, USA, Cat# MBS3805204) according to the manufacturer’s instructions. Briefly, 100 µL of plasma or urine samples, diluted as necessary, were added to wells coated with *SH3YL1*-specific antibodies and incubated for 2 h at room temperature. After washing, a biotinylated secondary antibody was added, followed by incubation with streptavidin-HRP for 30 min. The wells were then developed using a TMB substrate solution, and the reaction was stopped with a sulfuric acid solution. Absorbance was measured at 450 nm using a microplate reader, and *SH3YL1* concentrations were calculated by interpolation from a standard curve. All samples were run in duplicate to ensure accuracy, and results were expressed as mean concentrations (pg/mL or ng/mL).

### 4.4. Protein–Protein Interaction (PPI) Network Construction

To construct a protein–protein interaction (PPI) network, *SH3YL1* was uploaded to the STRING database (https://string-db.org/) using a medium confidence interaction score threshold of 0.400 [40]. Further protein–protein interaction analysis was conducted with Metascape (http://metascape.org) to explore functional annotations and pathway enrichment [41].

### 4.5. GO Analysis

The DAVID database platform (https://david.ncifcrf.gov) was used to perform GO analysis to study the biological functions of *SH3YL1*-associated genes [42]. A list of the *SH3YL1*-related key genes was uploaded, using the “official gene symbol” as the identifier. GO analysis was specifically used to screen for biological processes (BP), cellular components (CC), and molecular functions (MF). The top 10 enriched terms with the lowest *p*-values in each category were visualized using a GO Chord diagram and bubble charts. R version 4.4.3, RStudio (v2023.12.1+402), and the ggplot2 package (v3.5.1), with custom modifications to meet the needs of this study [43]. The GO Chord diagram was created to demonstrate the interrelation between *SH3YL1* and associated biological processes.

### 4.6. RNA Transcriptome Sequencing

Total RNA concentration was calculated by Quant-IT RiboGreen (Invitrogen, Carlsbad, CA, USA, #R11490). To assess the integrity of the total RNA, samples are run on the *TapeStation RNA screentape *(Agilent Technologies, Santa Clara, CA, USA, #5067-5576). Only high-quality RNA preparations, with RIN greater than 7.0, were used for RNA library construction. A library was independently prepared with 1 µg of total RNA for each sample by Illumina TruSeq Stranded mRNA Sample Prep Kit (Illumina, Inc., San Diego, CA, USA, #RS-122-2101). The first step in the workflow involves purifying the poly-A containing mRNA molecules using poly-T-attached magnetic beads. Following purification, the mRNA is fragmented into small pieces using divalent cations under elevated temperature. The cleaved RNA fragments are copied into first strand cDNA using SuperScript II reverse transcriptase (Invitrogen, #18064014) and random primers. This is followed by second strand cDNA synthesis using DNA Polymerase I, RNase H, and dUTP. These cDNA fragments then go through an end repair process, the addition of a single ‘A’ base, and then ligation of the adapters. The products are then purified and enriched with PCR to create the final cDNA library. The libraries were quantified using KAPA Library Quantification kits for Illumina Sequencing platforms according to the qPCR Quantification Protocol Guide (KAPA BIOSYSTEMS, Wilmington, MA, USA #KK4854) and qualified using the TapeStation D1000 ScreenTape (Agilent Technologies, # 5067-5582). Indexed libraries were then submitted to an Illumina NovaSeq 6000 (Illumina, Inc., San Diego, CA, USA), and the paired-end (2 × 100 bp) sequencing was performed by Macrogen Incorporated, Seoul, Republic of Korea.

### 4.7. Data Acquisition

The data used for survival analysis were obtained from the publicly available TCGA Pan-Cancer Atlas (BLCA cohort). The dataset includes clinical and transcriptomic information for 411 patients diagnosed with bladder cancer. Patients were classified into metastatic and non-metastatic groups based on their clinical status. Within each group, *SH3YL1* and *NOX4* expression levels were used to further categorize patients into high-expression (top 50%) and low-expression (bottom 50%) groups. The data were accessed via the Genomic Data Commons (GDC) portal (https://portal.gdc.cancer.gov) on [1 December 2024].

### 4.8. Survival Analysis

Kaplan–Meier survival curves were generated using GraphPad Prism 10.3.1 software to assess the prognostic significance of *SH3YL1* and *NOX4* expressions. Data from the TCGA Pan-Cancer Atlas (BLCA cohort) were utilized. Patients were divided into metastatic and non-metastatic groups and further categorized into high- (top 50%) and low- (bottom 50%) expression groups for *SH3YL1* and *NOX4* based on mRNA expression levels. Survival endpoints included overall survival (OS) and disease-specific survival (DSS). Statistical differences between survival curves were evaluated using the log-rank test, with *p*-values below 0.05 considered significant.

### 4.9. Statistical Analysis

All experimental data were expressed as mean ± standard deviation (mean ± SD). Statistical analyses were performed using GraphPad Prism 10.3.1 software. For comparisons between groups, parametric or non-parametric tests were applied based on the data characteristics. An unpaired two-tailed *t*-test was used for comparing two groups, while one-way analysis of variance (ANOVA) followed by Tukey’s post hoc test was applied for multiple group comparisons. Normality of data was assessed using the Shapiro–Wilk test, and homogeneity of variances was evaluated with Levene’s test. If the assumptions for parametric tests were not met, non-parametric tests such as the Mann–Whitney U test or Kruskal–Wallis test were employed. Correlation analyses between *SH3YL1* and *NOX* family genes were conducted using Spearman’s rank correlation due to the non-parametric nature of the data. ROC curve analysis was performed to evaluate the diagnostic performance of *SH3YL1* and *NOX4* in distinguishing between NMIBC and MIBC subtypes, with the AUC values used as a measure of predictive accuracy. Statistical significance was defined as *p*-values below 0.05.

## 5. Conclusions

We suggest that the *SH3YL1–NOX4* axis plays a critical role in the invasive phenotype of MIBC, distinguishing it from NMIBC at a molecular level and highlighting the need for targeted biomarkers in advanced bladder cancer. The apparent independence of cisplatin-induced AKI from *SH3YL1* expression in MIBC further raises important questions about the unique nephrotoxic mechanisms within the MIBC subset of bladder cancer patients. These findings suggest that *SH3YL1* may not serve as a universal AKI biomarker, especially in cases where intrinsic pathway activation mitigates further stress induction by agents like cisplatin. Future studies should focus on elucidating alternative molecular pathways involved in cisplatin-induced renal damage, which could inform the development of nephroprotective strategies that preserve cisplatin’s antitumor efficacy without compromising renal function. Identifying these pathways has the potential to pave the way for clinical advancements that enhance the therapeutic index of cisplatin in bladder cancer treatment.

## Figures and Tables

**Figure 1 ijms-26-03959-f001:**
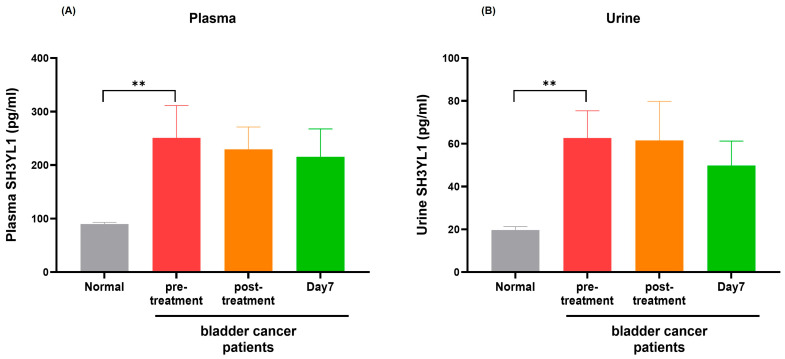
Quantitative analysis of *SH3YL1* levels in plasma and urine from MIBC patients. (**A**) Plasma *SH3YL1* levels in MIBC patients were measured before treatment, post-treatment, and on day 7, with results compared to health controls. *SH3YL1* levels were significantly elevated in MIBC compared to controls, without notable changes post-treatment. Results are presented as the mean ± SD. ** *p* < 0.01, indicating a significant difference from the control group. (**B**) Urine *SH3YL1* levels in MIBC demonstrated similar patterns, with significantly higher levels than in healthy controls. These findings suggest that *SH3YL1* levels remain consistently elevated in MIBC patients, irrespective of acute treatment effects, highlighting the potential of *SH3YL1* as a baseline biomarker in this patient population. Results are presented as the mean ± SD. ** *p* < 0.01, indicating a significant difference from the control group.

**Figure 2 ijms-26-03959-f002:**
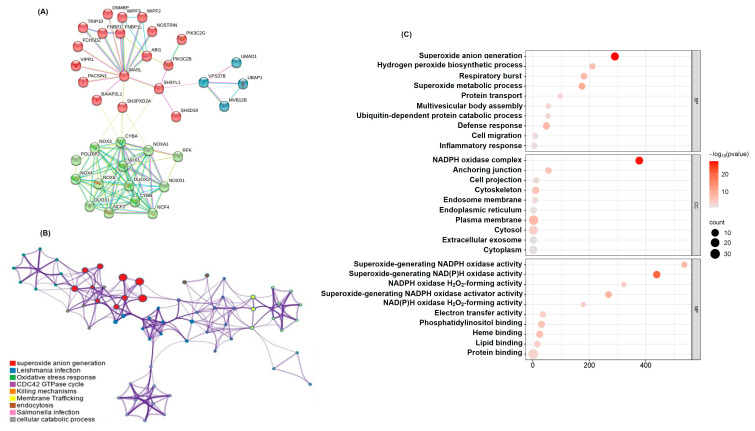
Functional network and pathway analysis of *SH3YL1*-associated genes in bladder cancer. (**A**) STRING database analysis identified 36 genes closely interacting with *SH3YL1*, categorized into three distinct clusters using k-means clustering based on interaction patterns and functional similarities. These clusters highlight associations relevant to bladder cancer pathology. (**B**) Metascape pathway and process enrichment analysis of *SH3YL1*-associated genes visualized as a network diagram, revealing significant enrichment in pathways such as oxidative stress response, actin cytoskeleton dynamics, and cell migration. (**C**) GO enrichment analysis displaying the top enriched terms across three categories—biological processes (BP), cellular components (CC), and molecular functions (MF). Each bubble represents an enriched term, with the size corresponding to the number of associated genes and the color intensity reflecting the −log10 (*p*-value).

**Figure 3 ijms-26-03959-f003:**
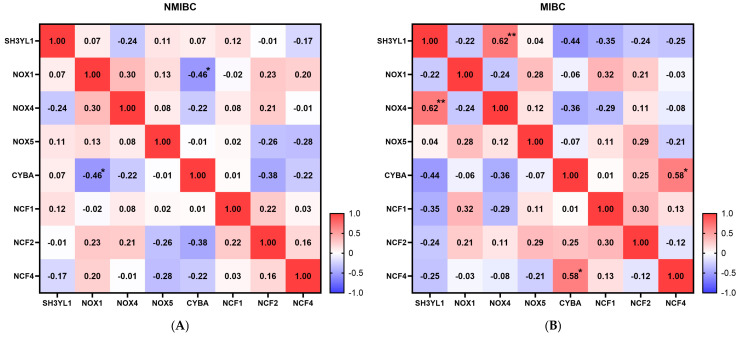
Correlation matrix of *SH3YL1* and NADPH oxidase-related genes in NMIBC and MIBC bladder tissue samples. (**A**) Correlation matrix for *SH3YL1* and NADPH oxidase-related genes (*NOX1*, *NOX4*, *NOX5*, *CYBA*, *NCF1*, *NCF2*, and *NCF4*) in NMIBC bladder tissue samples. Correlation coefficients are represented by color intensity, with red indicating positive correlations and blue indicating negative correlations. *SH3YL1* displays weak correlations with most NADPH oxidase-related genes in NMIBC samples, reflecting limited interaction within this subtype. (**B**) Correlation matrix for *SH3YL1* and NADPH oxidase-related genes in MIBC bladder tissue samples. A notable positive correlation is observed between *SH3YL1* and *NOX4* (r = 0.62, *p* < 0.01), suggesting a potentially significant interaction in MIBC. This higher correlation may indicate a coordinated expression pattern relevant to the aggressive phenotype of MIBC. Spearman’s correlation R-values are displayed, with statistically significant correlations indicated by asterisks (* *p* < 0.05, ** *p* < 0.01). The color intensity represents correlation coefficients, with a scale ranging from −1.0 (strong negative correlation) to 1.0 (strong positive correlation).

**Figure 4 ijms-26-03959-f004:**
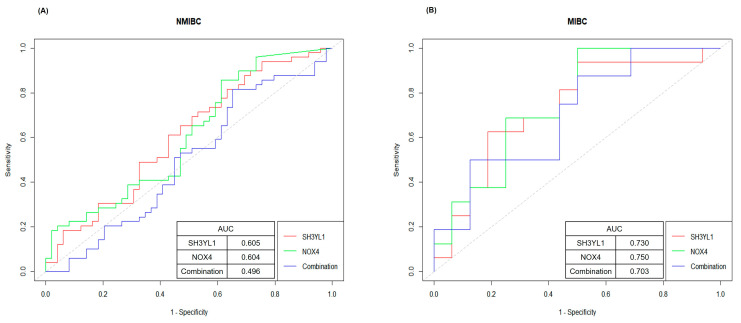
ROC curve analysis evaluating the predictive accuracy of *SH3YL1* and *NOX4* expression in NMIBC and MIBC bladder cancer subtypes. (**A**) ROC curve analysis for *SH3YL1*, *NOX4*, and their combination to assess their predictive accuracy in NMIBC samples. The AUC values for *SH3YL1*, *NOX4*, and the combination are 0.605, 0.604, and 0.496, respectively, indicating limited diagnostic performance in NMIBC. (**B**) ROC curve analysis for *SH3YL1*, *NOX4*, and their combination in MIBC samples, showing AUC values of 0.730 for *SH3YL1*, 0.750 for *NOX4*, and 0.703 for the combination. Both *SH3YL1* and *NOX4* exhibit strong predictive accuracy for MIBC.

**Figure 5 ijms-26-03959-f005:**
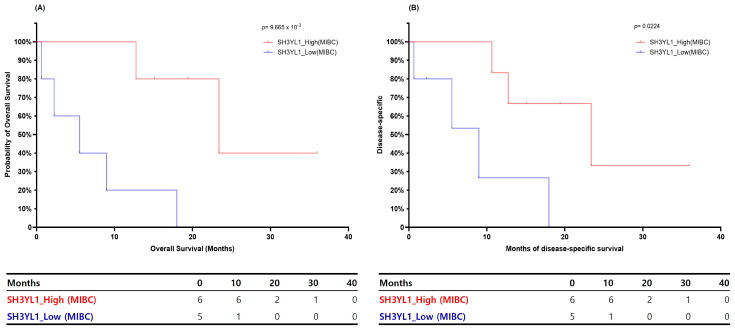
Kaplan–Meier survival analysis of *SH3YL1* expression in MIBC. (**A**) Overall survival (OS) and (**B**) disease-specific survival (DSS) curves for MIBC patients stratified by high and low *SH3YL1* expression levels. Patients with low *SH3YL1* expression exhibited significantly poorer survival in both OS (*p* = 9.665 × 10^−3^) and DSS (*p* = 0.0224), supporting its prognostic significance in muscle-invasive bladder cancer.

## Data Availability

The RNA-seq datasets used for survival analysis in this study are publicly available from The Cancer Genome Atlas (TCGA) at https://portal.gdc.cancer.gov/. The additional clinical and experimental data presented in this study are available on request from the corresponding author. These data are not publicly available due to privacy and ethical restrictions.

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
