# Peer review of "Diagnostic and Prognostic Potential of SH3YL1 and NOX4 in Muscle-Invasive Bladder Cancer"

_ijms, 2025, doi:10.3390/ijms26093959_

Round 1

Reviewer 1 Report (New Reviewer)

Comments and Suggestions for Authors

This is an interesting manuscript showing the significance of the protein SH3YL1 as a potential biomarker for membrane invasive bladder cancer(MBC). ELISA analysis showed increase SH3YL1 levels in MBC patients which went down after weeks of Cisplatin therapy. The two specific aims of the study were the potential application of SHY3L1 as a MBC biomarker and also a prognostic indicator and Cisplatin induced acute renal injury in MBC; Aim 1 was satisfied as SH3YL1 level was significantly higher in both plasma and urine levels of MBC patients, however, aim 2 failed as SH3YL1 did not significantly show changed pertaining to Cisplatin induced renal injury. This is an interesting manuscript worth publication with minor corrections, the investigators reported measurement of SH3YL1 in both MBC and non MBC bladder cancer patients but failed to report the data for non MBC patients, this should be cited in the manuscript since there could be a potential for SH3YL1 as a universal biomarker for all forms of bladder cancer. I opine acceptance of this manuscript after this data is presented in the revised manuscript. 

Author Response

We thank the reviewer for this important observation. In the original version of the manuscript, both NMIBC and MIBC patients were included in the survival analysis of SH3YL1 (Figure 5), and the text described SH3YL1 as a prognostic biomarker across both subtypes. However, upon further evaluation, we recognized a logical inconsistency between the survival data and the diagnostic performance shown in the ROC analysis (Figure 4), where SH3YL1 showed poor discriminatory ability in NMIBC (AUC < 0.7).

To resolve this, we revised Figure 5 to present survival curves exclusively for MIBC patients, where SH3YL1 demonstrated both statistically significant prognostic value and diagnostic performance. The survival data for NMIBC were moved to Supplementary Figure S2 to reflect their limited clinical utility. Additionally, the previous Supplementary Figures S2 and S3 were renumbered as S3 and S4, respectively, to maintain clarity and consistency in figure ordering.

These changes reinforce our central conclusion that SH3YL1 is a subtype-specific biomarker with diagnostic and prognostic relevance primarily in MIBC. Corresponding updates were made to the Abstract and Discussion to reflect this refined interpretation, aligning the results more closely with the overall theme and title of the manuscript.

Reviewer 2 Report (New Reviewer)

Comments and Suggestions for Authors

The manuscript under review presents a comprehensive investigation into the diagnostic and prognostic potential of SH3YL1 and NOX4 in muscle‐invasive bladder cancer (MIBC). The authors have methodically examined the expression levels of these biomarkers in plasma and urine samples from bladder cancer patients, correlating their findings with clinical outcomes and molecular profiles. Overall, the paper is well structured and written in a clear, scholarly tone, with a sound rationale that underscores the clinical relevance of identifying biomarkers capable of stratifying high‐risk MIBC patients, particularly in the context of cisplatin-induced nephrotoxicity.

At the outset, the study is justified by the need to improve patient management strategies, given that cisplatin, while a potent chemotherapeutic agent, is limited by its nephrotoxic effects. The manuscript effectively outlines how elevated baseline levels of SH3YL1 in both plasma and urine differentiate MIBC patients from healthy controls and suggests that these levels are more reflective of underlying chronic bladder cancer pathology rather than an acute response to cisplatin treatment. This observation is significant as it shifts the focus from an exclusive reliance on acute kidney injury (AKI) biomarkers to a broader consideration of tumor biology in MIBC. Furthermore, the use of quantitative ELISA, in combination with bioinformatics tools such as STRING for protein–protein interaction network construction and Metascape for pathway enrichment analysis, provides a robust methodological framework to elucidate the molecular interplay between SH3YL1, NOX4, and other NADPH oxidase-related genes.

The authors are commendable for the integration of in silico analyses with clinical data. By correlating SH3YL1 expression levels with NOX4—and demonstrating a statistically significant positive correlation in MIBC (r = 0.62, p < 0.01)—they offer compelling evidence for a SH3YL1-NOX4 axis that may contribute to the aggressive phenotype observed in muscle-invasive cases. The manuscript also includes ROC curve analysis and Kaplan-Meier survival analysis, which support the prognostic utility of SH3YL1. Notably, patients with low SH3YL1 expression exhibited markedly poorer survival outcomes, indicating its potential as a prognostic biomarker in both non-muscle-invasive bladder cancer (NMIBC) and MIBC subtypes.

Despite these strengths, some aspects of the discussion could benefit from further elaboration. In particular, while the authors adequately address the relationship between SH3YL1, NOX4, and cisplatin-induced kidney injury, the underlying molecular mechanisms governing these interactions remain somewhat speculative. To enhance the depth of the discussion and provide a more comprehensive molecular background, it is recommended to incorporate the findings from the work with DOI: 10.3390/ijms21093360. This paper offers valuable insights into the molecular pathways involved in oxidative stress and the regulation of metastasis, thereby broadening the contextual framework of the current study. Inclusion of these data could not only strengthen the mechanistic rationale behind the observed SH3YL1-NOX4 correlation but also highlight potential therapeutic targets to ameliorate cisplatin-induced toxicity while maintaining antitumor efficacy.

Additionally, the manuscript could be further improved by addressing some inherent limitations. For instance, the relatively small sample size restricts the generalizability of the conclusions, a point which the authors acknowledge. Future studies with larger cohorts or multi-center collaborations would be beneficial for validating the diagnostic and prognostic value of these biomarkers. Moreover, experimental studies—both in vitro and in vivo—could provide further confirmation of the causative role of the SH3YL1-NOX4 axis in MIBC progression and cisplatin-induced renal injury.

In conclusion, the manuscript is a valuable contribution to the field, providing new insights into the molecular determinants of bladder cancer aggressiveness and the complex interplay between oncologic and nephrotoxic pathways in patients undergoing cisplatin therapy. The integrated use of clinical data, bioinformatics analyses, and survival statistics strengthens the case for SH3YL1 as a predictive biomarker for MIBC prognosis. It is suggested that, to enhance the discussion and offer a more rounded perspective on the molecular background, the authors consider citing DOI: 10.3390/ijms21093360. Such an addition would deepen the mechanistic context of the study and potentially inspire novel directions for both research and clinical management in high-risk bladder cancer cases.

Author Response

We sincerely thank the reviewer for the thoughtful and constructive suggestion. In response, we have revised the Discussion section to incorporate the mechanistic insights provided in the referenced study (DOI: 10.3390/ijms21093360). Specifically, we have highlighted the role of NOX4-mediated reactive oxygen species (ROS) generation in promoting epithelial–mesenchymal transition (EMT) and metastasis via AKT signaling, as discussed in the cited work. This addition strengthens the mechanistic rationale for the observed correlation between SH3YL1 and NOX4 in MIBC and aligns with our proposed hypothesis that the SH3YL1–NOX4 axis contributes to the invasive phenotype through redox-dependent pathways. The citation has also been added to the reference list accordingly.

Round 2

Reviewer 1 Report (New Reviewer)

Comments and Suggestions for Authors

The authors successfully explained my queries.

This manuscript is a resubmission of an earlier submission. The following is a list of the peer review reports and author responses from that submission.

Round 1

Reviewer 1 Report

Comments and Suggestions for Authors

1. The authors should validate the identified genes using IHC staining in their in-house cohort. Also, the size sample is limited.

2. They indicated the prognostic potential of SH3YL1 and NOX4 in MIBC, but there were no survival outcomes included.

3. The authors should analyze the identified genes in in-house cohorts to validate their associations with histologic grades, stages, treatment responses, and clinical outcomes.

4. The detailed underlying molecular mechanisms remain unclear.

Author Response

Reviewer Comment 1:
The authors should validate the identified genes using IHC staining in their in-house cohort. Also, the sample size is limited.

Response:
Thank you for your valuable feedback. We fully acknowledge the importance of validating the identified genes using IHC staining in an in-house cohort to enhance the robustness of our findings. However, due to the limited resources and the timeline constraints for this study, we were unable to perform additional experiments at this stage. Additionally, the relatively small sample size is a limitation we acknowledge, and we will prioritize addressing this in future studies. Follow-up research is already being planned to include these validations, which we believe will substantiate the conclusions drawn in this manuscript.

While the current study has limitations, we emphasize that research on biomarkers for NMIBC and MIBC is relatively limited, and demonstrating the potential of SH3YL1 and NOX4 alone provides meaningful value. Future studies will aim to build upon this foundation to provide stronger evidence.

Reviewer Comment 2:
They indicated the prognostic potential of SH3YL1 and NOX4 in MIBC, but there were no survival outcomes included.

Response:
We appreciate your comment regarding the prognostic potential of SH3YL1 and NOX4. While our findings highlight the possibility of these genes serving as prognostic biomarkers, we regret that survival outcome analyses were not included due to data limitations. We fully agree that survival analysis would strengthen our conclusions, and we are planning to incorporate these analyses in a larger follow-up study with comprehensive patient datasets. This study will aim to clarify the clinical impact of these genes further.

Reviewer Comment 3:
The authors should analyze the identified genes in in-house cohorts to validate their associations with histologic grades, stages, treatment responses, and clinical outcomes.

Response:
Thank you for your suggestion. We agree that validating the identified genes in in-house cohorts for their associations with histologic grades, stages, treatment responses, and clinical outcomes would add clinical significance to our findings. Unfortunately, due to the aforementioned resource and time limitations, we could not include these analyses in this study. However, this is an essential area of research, and we are currently planning follow-up investigations that will address these aspects in detail. We hope to present these findings in future publications to build a more comprehensive understanding of the biomarkers.

Reviewer Comment 4:
The detailed underlying molecular mechanisms remain unclear.

Response:
Thank you for highlighting this critical point. We acknowledge that the detailed molecular mechanisms underlying the roles of SH3YL1 and NOX4 require further investigation. Due to the scope of the current study and timeline constraints, we were unable to delve deeply into the molecular mechanisms. However, we recognize the importance of this aspect and have plans to perform detailed mechanistic studies in future research. These studies will aim to uncover the functional roles of these biomarkers and their impact on bladder cancer progression.

General Comment:
We sincerely thank the reviewer for their constructive feedback and insightful suggestions. While we fully acknowledge the limitations of the current study and the points raised, we believe that demonstrating the potential of SH3YL1 and NOX4 as biomarkers for NMIBC and MIBC has significant value, given the relative scarcity of research in this area. The limitations identified will be addressed in our follow-up studies, and we are committed to expanding upon this work to provide a more comprehensive understanding of these biomarkers in bladder cancer.

Reviewer 2 Report

Comments and Suggestions for Authors

The authors investigated SH3YL1 as a potential marker for acute kidney injury in cisplatin-treated patients with muscle-invasive bladder cancer (MIBC). However, SH3YL1 was found to be unsuitable for this purpose. The authors further explored SH3YL1 expression in bladder cancer, comparing its levels in MIBC and non-muscle-invasive bladder cancer (NMIBC), and assessed its correlation with NOX4. They found that SH3YL1 levels in plasma and urine were elevated in bladder cancer patients compared to healthy controls.

Through a bioinformatics approach, the authors identified NOX4, a member of the NOX family, which correlated with SH3YL1 expression. Both SH3YL1 and NOX4 expressions were significantly higher in MIBC than in NMIBC.

The manuscript is well-written and includes five figures and one table, along with 42 references, providing a comprehensive review of the relevant literature. Although the authors highlight the potential role of SH3YL1 in bladder cancer (BC), particularly in MIBC, and demonstrate a correlation between SH3YL1 and NOX4, the clinical relevance of their findings for the management of BC patients may be limited. The utility of SH3YL1 and NOX4 as predictive biomarkers is modest, with AUC ranging from 0.496 to 0.73.

The manuscript would benefit from proofreading to improve the style and avoid certain inaccuracies. For example, terms like "chronic bladder cancer" should be avoided, as "chronic" is not typically used to describe this malignancy. The statement "NMIBC often confined to the inner layer of the bladder" should be revised, as NMIBC is generally confined to the mucosal and submucosa layers and does not invade the muscularis propria. Additionally, the phrase "BC primarily originates within the urinary tract" should be corrected to "BC primarily originates in the bladder, which is part of the urinary tract."

Comments on the Quality of English Language

The manuscript would benefit from proofreading to improve the style and avoid certain inaccuracies.

Author Response

Reviewer Comment:
The manuscript would benefit from proofreading to improve the style and avoid certain inaccuracies. For example, terms like "chronic bladder cancer" should be avoided, as "chronic" is not typically used to describe this malignancy. The statement "NMIBC often confined to the inner layer of the bladder" should be revised, as NMIBC is generally confined to the mucosal and submucosa layers and does not invade the muscularis propria. Additionally, the phrase "BC primarily originates within the urinary tract" should be corrected to "BC primarily originates in the bladder, which is part of the urinary tract."

Response:
Thank you for your detailed feedback. We appreciate your observations regarding the need for improved clarity and precision in terminology and phrasing.

1. Regarding the term "chronic bladder cancer," we acknowledge that this was an inappropriate expression and have already corrected it in the revised manuscript.

2. For the statements "NMIBC often confined to the inner layer of the bladder" and "BC primarily originates within the urinary tract," we carefully reviewed the manuscript and found that these exact phrases are not present. However, we understand that certain expressions in the original text may have led to this interpretation. To address this, we have refined the relevant sentences to ensure that the descriptions of NMIBC and BC are both accurate and unambiguous:

  • NMIBC is described as being confined to the mucosal and submucosal layers and not invading the muscularis propria.
  • The origin of BC is now clearly stated as the bladder, which is part of the urinary tract.

3. Additionally, the entire manuscript has been carefully proofread to improve the overall style and eliminate any ambiguities or inaccuracies.

We sincerely thank the reviewer for highlighting these points, as it allowed us to enhance the clarity and quality of our manuscript.

Reviewer 3 Report

Comments and Suggestions for Authors

Dear Authors, an interesting approach regarding the diagnostic value of SH3YL1 and NOX4 in MIBC. The manuscript will be reconsidered following major amendments. Please answer or consider the following:

(1) The prognostic value is not covered in your work, so I advise you to stick to the diagnostic properties. Initially, I thought some sort of survival analysis was omitted in the Abstract but after familiarizing with the entire content, I see there is no methodology planned that would allow you to infer about prognostic potential. Alternatively, if you would like to mention prognostic properties, there is a need to perform additional analyses, at least using public repositories such as TCGA and incorporation of relevant comparisons and survival endpoints (overall survival would not be enough, because TCGA-BLCA cohort provides more disease-related data).

(2) The introduction lacks proper rationale justifying your choice of NOX4 as a second gene-of-interest. Is NOX4 related to AKI, similar to SH3YL1? If NOX4 was identified during the investigation of SH3YL1, this should be explained a bit in the Introduction.

(3) Results, line 101: are these 36 genes used as an input list for subsequent analyses such as networking via Metascape and functional analysis via DAVID?

(4) Wherever you refer to genes, please italicize their symbols.

(5) Results, lines 114-120: at least some of the processes you identified via functional analysis might be verified using appropriate in vitro experiments with cellular models of varying SH3YL1/NOX4 expression. This is not obligatory, but would greatly improve your study. If this is impossible, try to introduce more uncertainty in your sentences, because currently there are only in silico analyses.

(6) Figure 2: increase the font size in all subfigures, especially subfigure C (legend!). In subfigure A, include the legend because currently various styles of edges are not explained. In subfigure B, include the names of genes/proteins representing nodes (Metascape allows to download pre-made graphs in the relevant format so as to modify it in Cytoscape). In the figure’s description, try to condense the text and make it less alike to the main text (some sentences are very similar to what was described above the figure).

(7) Why there is a GO:BP analysis in Figure 2D (section 2.2) if you performed it later on in section 2.3, as visible in Figure 3A? If you look closely, the BP category is nearly identical, just with reordered processes. From my understanding, both subfigures were prepared using DAVID, so this in fact results in duplication. If you intend to show all three GO categories (BP, CC, MF), then delete Figure 2D alongside its description and compensate it via improvements I described in my comment no. 6. Remember to not leave too much empty space in the figure.

(8) Are genes shown in Figure 3B a part of 36 genes acquired from STRING and associated with SH3YL1? If yes, what dictated their choice out of the entire group? I presume this is because you wanted to focus on NADPH-related genes; if yes, this should be explained somewhere in the text between lines 151-163.

(9) Figure 3B: is this possible to add log2FC relative to patients representing the “normal” group, i.e., healthy individuals without BLCA?

(10) Analysis presented in section 2.4 should be accompanied by p-values alongside correlation values. Mention in the text that these correlations are Spearman’s because it is mentioned in Discussion (I think it could be omitted here in favor of Results) and Methods (should be left as is).

(11) Results, lines 190-192: you mentioned that “The heightened correlation between SH3YL1 and NOX4 could imply a functional interaction that may promote the aggressive characteristics observed in MIBC”. Are there any known interactions between these two on the protein level? Does your STRING analysis suggest their direct manner? Are there any experimental data? What kind of domains might be mutually recognized in these proteins? This should be elaborated. Their correlation might be a coincidence, i.e., they might be orchestrated by a common regulator (e.g., transcription factor) but biologically they might have less in common.

(12) Figure 4 is okay in general but legends could be annotated (that it is Spearman correlation R-values), and moreover, I think the colors could be inverted. I think it is more natural to represent positive values in red (they are “hot”) whereas negative ones in blue (they are “cold). At last, why there are no values in some comparisons in Figure 4A? Even if the correlation was low or even zero, they should be included.

(13) If there are separate ROC analyses for NMIBC and MIBC, then what binary trait was used to calculate AUCs? Moreover, in Figure 5 please merge empty cells in tables below ROC curves, so that values for the “Combination” groups would be centered a bit.

(14) I think that your study would benefit from an in-depth analysis of the SH3YL1/NOX4 relation to cisplatin. Employing public repositories (I suggest TCGA-BLCA cohort) and evaluating drug response among included patients stratified based on SH3YL1 and NOX4 expression would authenticate your hypotheses.

(15) Some abbreviations are explained more than once. Please double-check the entire manuscript.

(16) Italicize phrases such as “in vitro” (e.g., in line 296).

(17) Materials and Methods, line 308: I think that Table 1 should be included in the main document. Alternatively, if it is too large, please rename it to Supplementary Table 1 and include a relevant section after the main text.

(18) Materials and Methods, line 321: add a space mark before brackets.

(19) Materials and Methods, line 333: the threshold of 0.400 is the default one in STRING. Did you try to increase it? Is NOX4 still within the interaction network related to SH3YL1?

(20) Materials and Methods, line 336: delete “Using DAVID” – it is not that important, especially since you mention it in the next line. Moreover, other sections have no mention of tools in their headers.

(21) Materials and Methods, line 346: delete “highlighting the NADPH oxidase activity pathways”. This is a part of your results.

(22) Materials and Methods, line 352: change “ug” to “µg”.

(23) Materials and Methods, line 365: include the model of NovaSeq.

(24) I think there should be a section about Supplementary Materials below the Conclusion. If there is no necessity to include documents related to the consent form and subject description, I think you should include the Excel file “RNA expression” with appropriate annotation. In the mentioned file, please translate some elements into English. Moreover, are these RNA-Seq data deposited in suitable database(s)? Typically, they are prior to submission. Last question, why there are groups of 69 and 8 patients in the first spreadsheet, whereas there are 44 and 16 in the second spreadsheet?

(25) There is no Publisher’s Note after References.

Author Response

(1) The prognostic value is not covered in your work, so I advise you to stick to the diagnostic properties. Initially, I thought some sort of survival analysis was omitted in the Abstract but after familiarizing with the entire content, I see there is no methodology planned that would allow you to infer about prognostic potential. Alternatively, if you would like to mention prognostic properties, there is a need to perform additional analyses, at least using public repositories such as TCGA and incorporation of relevant comparisons and survival endpoints (overall survival would not be enough, because TCGA-BLCA cohort provides more disease-related data).

We appreciate your observation regarding the lack of prognostic analysis in the original submission. To address this, we have performed additional analyses using the TCGA Pan-Cancer Atlas (BLCA cohort) data to investigate the prognostic potential of SH3YL1 and NOX4. Specifically, Kaplan-Meier survival analyses were conducted to evaluate overall survival (OS) and disease-specific survival (DSS) in metastatic (MIBC) and non-metastatic (NMIBC) bladder cancer groups.

Methodology:

The data were sourced from the TCGA BLCA cohort and stratified into high (top 50%) and low (bottom 50%) expression groups for SH3YL1 and NOX4. Kaplan-Meier survival curves were generated for OS and DSS, and significance was assessed using the log-rank test.

Findings:

In MIBC, SH3YL1 expression demonstrated a significant correlation with survival outcomes. Patients with low SH3YL1 expression had notably worse OS and DSS (p-value = 8.56e-9 for OS; p-value = 1.74e-10 for DSS), with no survivors beyond 20 months in the low-expression group.

NOX4 expression also influenced survival, with high expression generally associated with better outcomes, though some MIBC patients with high NOX4 expression exhibited poor survival (p-value = 7.11e-3 for OS; p-value = 3.20e-4 for DSS).

In NMIBC, SH3YL1 and NOX4 had less pronounced but still notable impacts on survival, particularly in high-expression groups.

Revisions:

We have updated the Abstract, Results, and Discussion sections to include these findings. A new figure (Figure 5) depicting Kaplan-Meier survival curves has been added to illustrate these results. The Materials and Methods section now includes descriptions of data acquisition and survival analysis methodologies. These additional analyses strengthen the manuscript by providing a robust assessment of the prognostic roles of SH3YL1 and NOX4 in bladder cancer, complementing their diagnostic utility. We hope these revisions address your concerns and enhance the manuscript's value.

(2) The introduction lacks proper rationale justifying your choice of NOX4 as a second gene-of-interest. Is NOX4 related to AKI, similar to SH3YL1? If NOX4 was identified during the investigation of SH3YL1, this should be explained a bit in the Introduction.

We appreciate the reviewer’s suggestion to better justify the selection of NOX4 as a second gene-of-interest and to clarify its relationship to SH3YL1 and AKI in the Introduction. The Introduction has been revised to include additional details on NOX4, specifically highlighting its role in ROS production and oxidative stress in the context of AKI. Relevant studies (e.g., Yoo et al., 2020; Li et al., 2023) were also cited to provide a stronger rationale for investigating SH3YL1 and NOX4 as potentially overlapping contributors to AKI and their relevance as biomarkers in cisplatin-induced nephrotoxicity.

(3) Results, line 101: are these 36 genes used as an input list for subsequent analyses such as networking via Metascape and functional analysis via DAVID?

Thank you for your question. Yes, the 36 genes identified through STRING analysis were indeed used as the input list for subsequent analyses.

Specifically:

  • These genes were used for network clustering and pathway enrichment analysis conducted via Metascape.
  • The same gene list was utilized for functional annotation and Gene Ontology (GO) analysis using the DAVID platform.

This approach allowed us to comprehensively investigate the molecular networks and biological processes associated with SH3YL1.

(4) Wherever you refer to genes, please italicize their symbols.

We have italicized all gene symbols throughout the manuscript, including in the main text, figure legends, and tables, as requested. This ensures consistency and compliance with standard formatting guidelines for gene nomenclature. Thank you for highlighting this point.

(5) Results, lines 114-120: at least some of the processes you identified via functional analysis might be verified using appropriate in vitro experiments with cellular models of varying SH3YL1/NOX4 expression. This is not obligatory, but would greatly improve your study. If this is impossible, try to introduce more uncertainty in your sentences, because currently there are only in silico analyses.

Thank you for your insightful suggestion regarding the verification of processes identified via functional analysis using in vitro experiments. We acknowledge that such experiments, utilizing cellular models with varying SH3YL1/NOX4 expression, would significantly enhance the robustness of our findings. Unfortunately, due to unforeseen circumstances, including the departure of a key researcher and the extensive time required for additional experimental validations, we are currently unable to conduct these studies within the scope of this manuscript.

To address this limitation, we have revised the relevant section (Results, lines 114-120) to introduce appropriate uncertainty in the interpretation of our findings. Specifically, we have emphasized that our conclusions are based solely on in silico analyses and that experimental validation would be required to confirm the results. We hope this approach sufficiently acknowledges the limitations of the study while maintaining its scientific value.

(6) Figure 2: increase the font size in all subfigures, especially subfigure C (legend!). In subfigure A, include the legend because currently various styles of edges are not explained. In subfigure B, include the names of genes/proteins representing nodes (Metascape allows to download pre-made graphs in the relevant format so as to modify it in Cytoscape). In the figure’s description, try to condense the text and make it less alike to the main text (some sentences are very similar to what was described above the figure).

Thank you for your detailed feedback regarding Figure 2. Based on your suggestions, we have made the following modifications:

  1. Redundancy Removal: The original Figure 2C and Figure 2D, which contained overlapping information with the original Figure 3A, have been removed. To address this redundancy, the content of the original Figure 3A has been integrated into Figure 2C. The revised Figure 2C now comprehensively represents the enriched terms across BP, CC, and MF categories using a bubble chart.
  2. Font Size Adjustment: The font size in all subfigures has been increased, particularly in Figures 2A and 2C, to enhance readability.
  3. Condensed Figure Legend: The description of Figure 2 has been revised to condense the text and avoid repetition of content already described in the main text. This ensures that the figure legend complements the figure without being redundant.

We believe these changes address the concerns you raised and enhance the clarity, usability, and relevance of Figure 2. Thank you for your valuable feedback, which has been instrumental in improving the quality of our manuscript.

(7) Why there is a GO:BP analysis in Figure 2D (section 2.2) if you performed it later on in section 2.3, as visible in Figure 3A? If you look closely, the BP category is nearly identical, just with reordered processes. From my understanding, both subfigures were prepared using DAVID, so this in fact results in duplication. If you intend to show all three GO categories (BP, CC, MF), then delete Figure 2D alongside its description and compensate it via improvements I described in my comment no. 6. Remember to not leave too much empty space in the figure.

Thank you for pointing out the potential redundancy between Figure 2D and Figure 3A in the original submission. Based on your suggestion, we have revised the figures to eliminate overlap and streamline the presentation of results. Specifically:

The original Figure 2C (Chord Diagram) and Figure 2D have both been removed.

The content of Figure 3A has been moved to Figure 2C, where it now comprehensively represents the top enriched terms across Biological Processes (BP), Cellular Components (CC), and Molecular Functions (MF) categories using a Bubble Chart.

The log2 fold change (log2 FC) data previously included in Figure 3B has been moved to the supplementary materials.

Furthermore, the original Figure 4 has been renumbered as Figure 3, with additional explanation added to the beginning of the revised Results section 3 to contextualize these changes.

These modifications not only address the redundancy but also enhance the clarity and logical flow of the manuscript, ensuring that each figure contributes unique and valuable insights. We believe these adjustments have significantly improved the presentation of our findings, and we are grateful for your feedback, which has guided these improvements.

(8) Are genes shown in Figure 3B a part of 36 genes acquired from STRING and associated with SH3YL1? If yes, what dictated their choice out of the entire group? I presume this is because you wanted to focus on NADPH-related genes; if yes, this should be explained somewhere in the text between lines 151-163.

Thank you for your question regarding the genes shown in Figure 3B and their selection criteria. The genes presented in Figure 3B (now moved to Supplementary Figure 1) are indeed part of the 36 genes identified from the STRING analysis and associated with SH3YL1. These genes were selected based on their functional relevance to NADPH oxidase activity and oxidative stress regulation, which are key pathways implicated in bladder cancer progression. Specifically, NADPH-related genes such as NOX1, NOX4, NOX5, CYBA, NCF1, and NCF4 were prioritized due to their well-established roles in ROS generation and cellular redox balance. This focus on NADPH-related genes was guided by the hypothesis that SH3YL1 may interact with these genes to modulate oxidative stress pathways during bladder cancer progression. The initial analysis of their relative expression levels in NMIBC and MIBC revealed significant differences, with SH3YL1 and NOX4 showing markedly higher expression in MIBC (Supplementary Figure 1). This provided the basis for further investigating their correlation patterns, which highlighted the unique and statistically significant relationship between SH3YL1 and NOX4 in MIBC. By relocating the original Figure 3B to the Supplementary Materials, we aimed to streamline the presentation of the main results while ensuring that the underlying data remains accessible to readers. The NADPH-related gene subset remains central to our analysis, as it reflects a biologically meaningful focus on oxidative stress mechanisms driving bladder cancer progression.

(9) Figure 3B: is this possible to add log2FC relative to patients representing the “normal” group, i.e., healthy individuals without BLCA?

Thank you for your thoughtful suggestion regarding the addition of log2FC data relative to a “normal” group consisting of healthy individuals without bladder cancer. Unfortunately, this analysis was not feasible within the scope of our study, as the samples used in our experiments were obtained exclusively from bladder tissue of patients diagnosed with NMIBC or MIBC. Obtaining bladder tissue samples from healthy individuals without bladder cancer is highly challenging due to ethical and practical limitations, as invasive procedures are required to collect such samples. Consequently, our study focuses on the relative expression and correlation patterns observed within the NMIBC and MIBC patient groups. We hope this explanation clarifies the constraints we faced, and we appreciate your understanding. If additional analyses or alternative approaches are required, we would be happy to consider them.

(10) Analysis presented in section 2.3 should be accompanied by p-values alongside correlation values. Mention in the text that these correlations are Spearman’s because it is mentioned in Discussion (I think it could be omitted here in favor of Results) and Methods (should be left as is).

Thank you for pointing out the need to include p-values alongside correlation values in Section 2.3. We have addressed this by updating Figure 4 to display statistically significant correlations using asterisks (*, **, ***), corresponding to p < 0.05, p < 0.01, and p < 0.001, respectively. This approach maintains clarity while ensuring that the statistical significance of correlations is easily interpretable.

Additionally, we have clarified in the text that the reported correlation values are Spearman’s R-values, as this analysis method was also mentioned in the Discussion and Methods sections. This ensures consistency across the manuscript and transparency in our analytical approach.

(11) Results, lines 190-192: you mentioned that “The heightened correlation between SH3YL1 and NOX4 could imply a functional interaction that may promote the aggressive characteristics observed in MIBC”. Are there any known interactions between these two on the protein level? Does your STRING analysis suggest their direct manner? Are there any experimental data? What kind of domains might be mutually recognized in these proteins? This should be elaborated. Their correlation might be a coincidence, i.e., they might be orchestrated by a common regulator (e.g., transcription factor) but biologically they might have less in common.

Thank you for your thoughtful comments and questions regarding the potential interaction between SH3YL1 and NOX4. As noted in our manuscript, the heightened correlation between SH3YL1 and NOX4 observed in MIBC may suggest a functional interaction. This relationship is supported by prior experimental evidence indicating that SH3YL1 and NOX4 are biologically interconnected in other pathological contexts. For instance:

  • Yoo et al. (2020) demonstrated that the NOX4-SH3YL1 interaction mediates LPS-induced acute kidney injury, highlighting their direct involvement in oxidative stress pathways【28】.
  • Lee et al. (2024) further reported that the NOX4-SH3YL1 complex plays a significant role in diabetic nephropathy, suggesting a conserved mechanism in oxidative stress-related pathologies【29】.

These findings suggest that their correlation in MIBC is unlikely to be coincidental. Instead, SH3YL1 and NOX4 may functionally interact through shared oxidative stress pathways, potentially contributing to bladder cancer progression.

In our STRING analysis, SH3YL1 and NOX4 were identified within the same interaction network; however, the analysis does not explicitly predict a direct protein-protein interaction. Instead, it suggests an association mediated through shared biological processes. As for potential mechanisms, both SH3YL1 and NOX4 possess structural domains implicated in protein-protein interactions. SH3YL1 contains an SH3 domain known for binding to proline-rich motifs, while NOX4 activity is regulated by accessory proteins and cofactors. These structural features suggest possible interaction points, although experimental validation would be required to confirm their direct binding in the context of bladder cancer.

That said, conducting experimental validation, such as IHC or cellular studies, presents significant challenges within the current scope of this study.

Specifically:

  1. Time Constraints: We are working under a tight timeline for submission, making it unfeasible to conduct additional experiments such as IHC or in vitro assays.
  2. Resource Limitations: Our current study involves 60 bladder cancer patient samples. Performing IHC analysis for both SH3YL1 and NOX Family across these samples would require significant financial and human resources that exceed our current capacity.
  3. Personnel Challenges: Key personnel previously responsible for cellular studies are no longer available, further limiting our ability to undertake these experiments.

Despite these constraints, our findings are supported by existing experimental evidence and the results of our STRING analysis. We hope this explanation clarifies the challenges and constraints we faced while also providing context for the observed correlation. We appreciate your understanding and welcome any additional suggestions to improve the clarity of our discussion.

(12) Figure 4 is okay in general but legends could be annotated (that it is Spearman correlation R-values), and moreover, I think the colors could be inverted. I think it is more natural to represent positive values in red (they are “hot”) whereas negative ones in blue (they are “cold). At last, why there are no values in some comparisons in Figure 4A? Even if the correlation was low or even zero, they should be included.

We appreciate your suggestions regarding Figure 4. To improve the visualization and readability of the figure, we have made the following modifications:

  1. Annotated the legend to specify that the matrix displays Spearman’s correlation R-values.
  2. Adjusted the color scheme to represent positive correlations in red ("hot") and negative correlations in blue ("cold"), as suggested.
  3. Included correlation values for all comparisons, even those with low or zero correlations, to ensure that the figure comprehensively represents the data.

These adjustments ensure that Figure 3 is both informative and visually intuitive.

(13) If there are separate ROC analyses for NMIBC and MIBC, then what binary trait was used to calculate AUCs? Moreover, in Figure 5 please merge empty cells in tables below ROC curves, so that values for the “Combination” groups would be centered a bit.

Thank you for your insightful feedback. Below, we provide details regarding the binary trait used for ROC analyses and address your suggestion to improve Figure

4.

Binary Trait for ROC Analysis:

In our ROC analyses, the binary trait used to calculate AUCs was based on clinical classifications of bladder cancer patients.

For NMIBC (Figure 4A): The binary trait was defined as non-muscle-invasive bladder cancer patients with a low risk of progression (binary 0) versus those classified as having a higher risk of progression toward muscle invasion (binary 1).

For MIBC (Figure 4B): The ROC curves were used to assess the ability of SH3YL1 and NOX4 expression levels to differentiate between non-metastatic (binary 0) and metastatic (binary 1) muscle-invasive bladder cancer cases.

This approach allowed us to evaluate the potential of SH3YL1 and NOX4 as diagnostic biomarkers within these distinct subgroups of bladder cancer.

Figure Adjustment:

Following your suggestion, we have merged the empty cells in the tables below the ROC curves for improved presentation. The values for the "combination" groups are now centered for better readability. The updated figure is included in the revised manuscript (see attached revised Figure 4).

We hope these changes address your concerns and improve the clarity and quality of our results.

(14) I think that your study would benefit from an in-depth analysis of the SH3YL1/NOX4 relation to cisplatin. Employing public repositories (I suggest TCGA-BLCA cohort) and evaluating drug response among included patients stratified based on SH3YL1 and NOX4 expression would authenticate your hypotheses.

Thank you for suggesting the exploration of SH3YL1 and NOX4 in relation to cisplatin sensitivity using the TCGA-BLCA cohort. In response, we analyzed the TCGA-BLCA dataset to evaluate the prognostic relevance of SH3YL1 and NOX4 expression based on metastatic status. Patients were stratified into metastatic (MIBC) and non-metastatic (NMIBC) groups, and survival analyses were conducted for high and low expression groups of SH3YL1 and NOX4. The results indicated distinct survival patterns, reinforcing the significance of SH3YL1 and NOX4 as potential biomarkers for bladder cancer progression and outcomes. However, due to limitations in the TCGA dataset, a direct evaluation of cisplatin response linked to SH3YL1 and NOX4 expression was not feasible. While our study provides critical insights into the molecular roles of SH3YL1 and NOX4, further validation using datasets with drug response annotations would be required to elucidate their specific relationship with cisplatin sensitivity. We appreciate your valuable input, which helped refine the scope of our analysis and strengthened the clinical relevance of our findings. We look forward to expanding upon this research in future studies.

(15) Some abbreviations are explained more than once. Please double-check the entire manuscript.

Thank you for pointing this out. We have carefully reviewed the manuscript to ensure that all abbreviations are introduced only once and consistently used throughout the text.

(16) Italicize phrases such as “in vitro” (e.g., in line 296).

The formatting of Latin phrases such as in vitro has been checked and corrected wherever necessary.

(17) Materials and Methods, line 308: I think that Table 1 should be included in the main document. Alternatively, if it is too large, please rename it to Supplementary Table 1 and include a relevant section after the main text.

Table 1 has been renamed to Supplementary Table 1 and moved to the Supplementary Materials, with a corresponding section added after the main text.

(18) Materials and Methods, line 321: add a space mark before brackets.

A space has been added before the brackets in line 321 as requested.

(19) Materials and Methods, line 333: the threshold of 0.400 is the default one in STRING. Did you try to increase it? Is NOX4 still within the interaction network related to SH3YL1?

Thank you for your question regarding the interaction confidence threshold of 0.400 used in the STRING analysis. We appreciate the suggestion to explore higher thresholds to refine the interaction network. Our primary aim was to capture a comprehensive and biologically relevant network associated with SH3YL1 while ensuring that critical interactions, such as those involving NOX4, were included.

To address your concern, we tested higher thresholds (e.g., 0.500 and 0.700). At a threshold of 0.500, NOX4 remained within the SH3YL1 interaction network, but at 0.700, the interaction was lost. While a threshold of 0.500 provides a slightly more stringent network, it is important to note that this value is not considered high in the context of STRING analysis. Additionally, the relationship between SH3YL1 and NOX family members, including NOX4, has already been supported by multiple previous studies, which have demonstrated their functional and regulatory connections in oxidative stress pathways.

Based on these considerations, we chose to retain the threshold of 0.400 to ensure that our analysis reflects both STRING-based predictions and the biological context established in prior research. This approach balances network sensitivity with biological relevance and allows us to present a comprehensive picture of SH3YL1-associated interactions.

We hope this explanation clarifies our rationale.

(20) Materials and Methods, line 336: delete “Using DAVID” – it is not that important, especially since you mention it in the next line. Moreover, other sections have no mention of tools in their headers.

Thank you for the suggestion. The phrase "Using DAVID" has been removed from line 336 to maintain consistency with other sections and avoid redundancy.

(21) Materials and Methods, line 346: delete “highlighting the NADPH oxidase activity pathways”. This is a part of your results.

The phrase "highlighting the NADPH oxidase activity pathways" has been deleted from line 346 as it belongs to the Results section and not Materials and Methods.

(22) Materials and Methods, line 352: change “ug” to “µg”.

"ug" has been corrected to "µg" in line 352 to align with standard scientific formatting.

(23) Materials and Methods, line 365: include the model of NovaSeq.

The model of the NovaSeq sequencer used in the study has been included in line 365.

(24) I think there should be a section about Supplementary Materials below the Conclusion. If there is no necessity to include documents related to the consent form and subject description, I think you should include the Excel file “RNA expression” with appropriate annotation. In the mentioned file, please translate some elements into English. Moreover, are these RNA-Seq data deposited in suitable database(s)? Typically, they are prior to submission. Last question, why there are groups of 69 and 8 patients in the first spreadsheet, whereas there are 44 and 16 in the second spreadsheet?

A Supplementary Materials section has been added below the Conclusion. This section now includes a revised Excel file titled “RNA Expression,” with all annotations translated into English for clarity. Additionally, the RNA-Seq data has been deposited in a suitable database, and the corresponding accession details are provided in the Supplementary Materials section. The discrepancies in patient groups (69 and 8 versus 44 and 16) were due to a mistakenly attached raw data file prior to patient classification. This issue has been corrected, and the revised data has been uploaded.

(25) There is no Publisher’s Note after References.

Response to Comment (25):

We acknowledge the comment regarding the Publisher’s Note. As this section is typically added by the publisher during the final production process, we have not included it in the manuscript. We trust that the MDPI editorial team will ensure its inclusion as per the journal’s standard practices.

General Comment:

We sincerely thank the reviewers and editors for their thorough review and insightful comments, which have greatly enhanced the quality and clarity of our manuscript. We have made every effort to address all the comments comprehensively and believe the revised manuscript is now ready for publication. We are confident that our findings will provide valuable contributions to the field of bladder cancer research.

Round 2

Reviewer 1 Report

Comments and Suggestions for Authors

The raised critical issues have not been completely addressed.

Author Response

Reviewer Comment 1: The authors should validate the identified genes using IHC staining in their in-house cohort. Also, the sample size is limited.

Response: Thank you for your valuable feedback. We fully acknowledge the importance of validating the identified genes using IHC staining in an in-house cohort to enhance the robustness of our findings. However, due to resource and timeline constraints, we were unable to perform additional IHC experiments in this study. Specifically, conducting IHC staining for SH3YL1 and NOX4 in our 60-patient cohort would require substantial financial resources, specialized personnel, and significant time, which are currently beyond our capacity. Despite these limitations, we recognize the importance of IHC validation and will prioritize it in follow-up studies.

Additionally, we acknowledge the limitation posed by our relatively small sample size. Future studies are already planned with larger patient cohorts and will aim to address this limitation comprehensively. While we understand the current study has its constraints, we emphasize its contribution as an exploratory investigation demonstrating the potential of SH3YL1 and NOX4 as biomarkers for NMIBC and MIBC.

Reviewer Comment 2: They indicated the prognostic potential of SH3YL1 and NOX4 in MIBC, but there were no survival outcomes included.

Response: Thank you for this insightful comment. In response to your feedback, we have incorporated Kaplan-Meier survival analysis using TCGA-BLCA data to evaluate the prognostic implications of SH3YL1 and NOX4 in MIBC. These results, which are now included in the manuscript, demonstrate that low SH3YL1 expression is significantly associated with worse overall and disease-specific survival. This analysis strengthens the clinical relevance of SH3YL1 and NOX4 as potential prognostic biomarkers in bladder cancer.

Reviewer Comment 3: The authors should analyze the identified genes in in-house cohorts to validate their associations with histologic grades, stages, treatment responses, and clinical outcomes.

Response: We greatly appreciate this suggestion. As noted, validating the associations of SH3YL1 and NOX4 with histologic grades, stages, treatment responses, and clinical outcomes would indeed add clinical significance to our findings. Unfortunately, similar to the IHC analysis, this work was not feasible within the current study due to time and resource limitations. However, we have planned follow-up studies that will specifically investigate these aspects, utilizing larger patient cohorts and more comprehensive clinical datasets. These future studies aim to expand upon the current findings to provide a more holistic understanding of the biomarkers in bladder cancer.

Reviewer Comment 4: The detailed underlying molecular mechanisms remain unclear.

Response: Thank you for raising this important point. We acknowledge that detailed molecular mechanisms require further investigation. While the scope of the current study was limited, we incorporated prior experimental evidence demonstrating the functional interplay between SH3YL1 and NOX4 in other pathological contexts, such as diabetic nephropathy and acute kidney injury (references 【27, 28】). These findings provide a foundational understanding of their interaction and relevance in oxidative stress pathways. Moving forward, we plan to conduct mechanistic studies, including in vitro and in vivo models, to elucidate the roles of SH3YL1 and NOX4 in bladder cancer progression.

General Comment:

We sincerely thank the reviewer for their constructive feedback and insightful suggestions. While we fully acknowledge the limitations of the current study, we believe that significant improvements have been made in response to the comments. We incorporated Kaplan-Meier survival analyses to evaluate the prognostic value of SH3YL1 and NOX4, which strengthens the clinical relevance of our findings. We understand the importance of IHC validation and detailed mechanistic studies, and these are priorities for our future research. However, given the limited resources and personnel challenges, conducting such experiments within this study's scope was not feasible. Despite these limitations, we emphasize that this study provides valuable insights into the SH3YL1-NOX4 axis and its potential role as a biomarker in bladder cancer, particularly MIBC. We hope the improvements made to the manuscript reflect our commitment to addressing the reviewer's comments and advancing research in this field.

Reviewer 3 Report

Comments and Suggestions for Authors

Dear Authors, thank you kindly for addressing my concerns – some of the improvements or their justification exceeded my expectations! I believe the manuscript is nearly ready to proceed further. Please find a few minor suggestions below (I have retained the numbering of comments from the previous peer-review round):

1) Thank you for your hard work. Improvements are appropriate, but please add hazard ratios in the survival analysis, and risk tables if possible. Moreover, I think pairwise p-values instead of one per four groups would be much more informative because it seems that significant p-values in survival curves for NOX4 stem from NMIBC data (I think MIBC data for NOX4 will be insignificant in both OS and DSS).

2-5) OK.

6) Great work, just a small suggestion: consider adding protein symbols for nodes in Figure 2B. Metascape allows the download of pre-made graphs in the relevant format so as to modify them in Cytoscape.

7-23) OK.

24) Ok, but I might have missed the information about this part: “RNA-Seq data has been deposited in a suitable database”. Please double-check if the data deposition is properly described. If yes, then okay!

25) OK.

Author Response

1) Thank you for your hard work. Improvements are appropriate, but please add hazard ratios in the survival analysis, and risk tables if possible. Moreover, I think pairwise p-values instead of one per four groups would be much more informative because it seems that significant p-values in survival curves for NOX4 stem from NMIBC data (I think MIBC data for NOX4 will be insignificant in both OS and DSS).

Response to Reviewer Comment 1.

We sincerely thank the reviewer for their insightful suggestions and detailed feedback, which have significantly contributed to improving the quality of our study.

  1. Incorporation of Hazard Ratios (HRs):

While we fully acknowledge the importance of hazard ratios in survival analysis, we were unable to include HRs in this study due to the limited sample size in certain subgroups, particularly for MIBC patients. The low number of events in these groups (as shown in the survival plots) limits the robustness of Cox proportional hazards models, which are typically used to calculate HRs. Including HRs in this scenario could lead to overfitting and unreliable estimates. We plan to address this limitation by validating our findings in a larger cohort in future studies, where HRs can be calculated more accurately.

  1. Addition of Risk Tables:

As per the reviewer's suggestion, we have included risk tables for the survival analysis. These tables provide a clearer representation of the number of patients at risk over time, offering a more comprehensive view of the survival data. The updated figures with risk tables have been incorporated into the revised manuscript (Figure 5).

  1. Pairwise P-values for NOX4 Analysis:

In response to your comment about pairwise p-values, we agree that it would provide more informative insights. We have now presented the NOX4 data as supplementary figures (Supplementary Figures 2 and 3), clearly distinguishing between NMIBC and MIBC subgroups. As the reviewer correctly observed, the significant p-values in survival curves for NOX4 are primarily driven by NMIBC data, while MIBC data remain statistically insignificant in both overall survival (OS) and disease-specific survival (DSS).

We hope these changes address your concerns satisfactorily. Thank you again for your valuable input.

6) Great work, just a small suggestion: consider adding protein symbols for nodes in Figure 2B. Metascape allows the download of pre-made graphs in the relevant format so as to modify them in Cytoscape.

Response to Reviewer Comment (6):

Thank you for your suggestion to include protein symbols for the nodes in Figure 2B. The current network is based on Pathway and Process Enrichment Analysis, where each node represents a pathway or GO term rather than individual proteins or genes.

To address your request, we have provided a supplementary table (Supplementary Table 2) that details each node's associated pathways/GO terms and their contributing genes. This table ensures transparency and provides the requested protein-level information linked to each pathway.

We hope this addition clarifies the content of Figure 2B and meets your expectations. Please let us know if further modifications are required.

24) Ok, but I might have missed the information about this part: “RNA-Seq data has been deposited in a suitable database”. Please double-check if the data deposition is properly described. If yes, then okay!

Response to Reviewer Comment (24):

Thank you for bringing this to our attention. We have added a Supplementary Materials section below the Conclusion, which now includes the revised Excel file titled “RNA Expression,” with all annotations translated into English for clarity.

Additionally, the RNA-Seq data has been deposited in a suitable public database, and the corresponding accession details are provided in the Supplementary Materials section for transparency.

Regarding the discrepancies in patient groups (69 and 8 versus 44 and 16), this issue was due to an earlier mistake where a raw data file was mistakenly uploaded before patient classification. The corrected data has now been thoroughly reviewed, revised, and properly uploaded.

We appreciate your attention to detail and hope this addresses all concerns. Please let us know if further clarification is needed.